# Detection of a Diverse Endophyte Assemblage within Fungal Communities Associated with the Arundo Leaf Miner, *Lasioptera donacis* (Diptera: Cecidomyiidae)

**Marie-Claude Bon** [1,*], **John A. Goolsby** [2], **Guy Mercadier** [1], **Fatiha Guermache** [1], **Javid Kashefi** [1], **Massimo Cristofaro** [3], **Ann T. Vacek** [4] and **Alan Kirk** [1]

1   European Biological Control Laboratory, USDA-ARS, 34980 Montferrier-sur-Lez, France; jkashefi@ars-ebcl.org (J.K.)
2   Cattle Fever Tick Research Laboratory, USDA-ARS, Edinburg, TX 78541, USA; john.goolsby@usda.gov
3   Biotechnology and Biological Control Agency (BBCA) Onlus, 00123 Rome, Italy; m.cristofaro55@gmail.com
4   Department of Biology and School of Earth, Environment and Marine Sciences, The University of Texas-Rio Grande Valley, Edinburg, TX 78539, USA
*   Correspondence: mcbon@ars-ebcl.org; Tel.: +33-4-99-62-30-41

**Abstract:** The larvae of *Lasioptera donacis* Coutin feed on fungal communities lining galleries within the mesophyll of leaf sheaths of *Arundo donax* in an aggregative manner. It has been stated that *L. donacis* could have established a fundamental symbiotic relationship with one fungus, although the fungal composition of these communities remains unsettled. Using a culture-dependent approach and ITS sequencing, the present work characterizes and compares the fungal communities associated with *L. donacis* in Eurasia with the endophytes of *A. donax* in Texas where *L. donacis* is absent. The 65 cultivable isolates obtained from *L. donacis* fungal communities were sorted into 15 MOTUs, among which *Fusarium* and *Sarocladium* predominated. No particular MOTU was systematically recovered from these communities regardless of the sites. The 19 isolates obtained in Texas were sorted into 11 MOTUs. *Sarocladium* and *Fusarium* were commonly found in Texas and Eurasia. Our finding indicate that the communities were composed of a diverse assemblage of non-systemic endophytes, rather than an exclusive fungal symbiont. From ovipositors and ovarioles of *L. donacis* emerging from plants in France, we opportunistically isolated the endophyte *Apiospora arundinis*, which lies at the origin of further research pertaining to its role in the feeding and oviposition of *L. donacis*.

**Keywords:** fungal community; Cecidomyiidae; endophytes; grass; saprobes

## 1. Introduction

*Arundo donax* L. (Poales: Poaceae), commonly named giant reed is a highly invasive, bamboo-like weed considered a severe threat to riparian habitats throughout the southern half of the United States, where it causes erosion [1] reduces riparian biodiversity [2], and consumes excessive amounts of water [3,4]. A program of classical biological control regarded as a cost-effective and sustainable option for the management of the weed over large areas, such as the Rio Grande was undertaken in the early 2000s [5]. In 2016, a midge, *Lasioptera donacis* Coutin and Faivre-Amiot, 1981 (Diptera: Cecidomyiidae), was approved for release in the United States and Mexico [6]. In its southwestern European native range, *L. donacis* larvae develop galleries inside the mesophyll of leaf sheaths of the giant reed. Their feeding leads to premature defoliation increasing light penetration through the canopy, which is expected to accelerate the recovery of the native riparian plant community following introduction of this agent [6,7]. *L. donacis* belongs to the Cecidomyiidae family, which represents the largest radiation of gall-forming insects, characterized by complex trophic interactions among plants, fungal symbionts, parasites, and predators [8,9]. Of all genera among this family, *Lasioptera* Meigen, 1818 is arguably the most poorly understood

although it includes not less than 129 species distributed in the Old World, Australia, and America [10].

The genus *Lasioptera* was defined taxonomically by the presence on the postabdomen and ovipositor of adult females of mycangia structures, which function in the transport of fungal conidia [10]. Most *Lasioptera* species are gall inducers, particularly in stems, but others are either inquilines in galls of other gall midges or induce swellings on stems to develop larvae without making galls [8,10–12]. The galls or the swollen stems produced by the *Lasioptera* species house a fungal mat, historically termed as Ambrosia [8,13–16]. As Ambrosia fungi are known to be associated with certain members of beetles of the subfamilies of Scolytinae and Platypodinae (Coleoptera: Curculionidae), but not with Dipterans to our knowledge [17], we would refer to fungal communities when it comes to fungi that either directly or indirectly mediate the nutritional interaction of midge larvae with host plants [8,13–16]. In the most-documented cases, the gall midge species has specialized relationships with one fungal species [16–18]. In *Lasioptera*-induced galls, the only fungi found so far were *Sporothrix* sp., *Ramichloridum subulatum*, and *Aureobasidium pullulans* as reviewed by Rohfritsch [14–16]. The association between *Lasioptera arundinis*, which feeds on common reed (*Phragmites australis* (Cav.) Streudel, and *Ramichloridium subulatum* was relatively well studied and described as a true mutualism [16]. The fungus provides nutrition to the gall midge larvae and by inducing lysis of stem cells opens a channel to the vascular bundles. When larvae stop feeding in the mature galls, the fungus proliferates. In contrast, fungus acquisition and deposition by *Lasioptera* mining midges are less well known. According to Coutin [19], *L. donacis* larvae might be sapromycophagous, feeding on a black mold which grows in old galleries in the leaf sheath. These old galleries where *L. donacis* larvae would live had been bored by a leaf miner fly, *Cerodontha phragmitophila* Hering [19], though this observation was not confirmed by [20]. Recently, Careddu et al. [21] evidenced through a stable isotope analysis for C and N a direct trophic interaction between the *L. donacis* larvae, the fungal communities, and the giant reed mesophyll, thereby positioning *L. donacis* as an omnivore.

In his description of the species, Coutin [19] by observing that the larvae feed in an aggregative manner on the mycelia of fungi, inferred a symbiotic association between the *L. donacis* larvae and the fungi forming the black mold, and put forward *Aspergillus niger* Tiegh as one primary fungus. Other than this assumption, no information per se exists regarding the accurate characterization of the primary fungus, its occurrence over a large biogeographic region, or the overall composition of the fungal communities, and this lack of information provided the impetus for the present research.

In this study, our main purpose was to clear up the uncertainty regarding Coutin's identification of the primary fungus constituted of the black mold developing inside *Lasioptera* galleries, corresponding to the later stages of the fungal communities, from across much of the Mediterranean Latin Arch, from Spain to Greece. An additional purpose of this research was to characterize the cultivable assemblage of endophytes associated with leaf sheaths of *A. donax* in Texas where the midge is intended for release. During investigations into the characterization of these fungal communities, conidiae carried by the ovipositors or located near the ovarioles of the midge were opportunistically isolated. Therefore, an additional purpose sought to identify these conidiae. We purposely chose to use a traditional approach dependent on culture, and identification based on molecular markers for three reasons, i.e., the expectation of isolating and culturing the primary fungal symbiont suggested by [19], the lack of taxonomic expertise from local mycologists, and the need to speed up this analysis to be used in a more comprehensive study. We targeted the nuclear ribosomal DNA (nrDNA) internal transcribed spacer (ITS) which has been widely used in both molecular systematics and ecological studies of fungi [22–24] and has been selected as the formal barcode marker for fungi [23]. Because of its multicopy nature, the ITS allows easy amplification from samples containing low DNA concentrations. Furthermore, thousands of ITS sequences of different species are readily available from various online

databases including UNITE [25] and the International Nucleotide Sequence Databases such as GenBank, providing a large reference collection for taxonomic classification.

## 2. Materials and Methods

### 2.1. Fungal Communities Associated with L. donacis

A total of 24 populations of *A. donax* were surveyed in 2011, 2012, and 2013 in *L. donacis'* native range, along the Mediterranean Latin Arch from Spain to Greece (Table S1).

In each population, we identified 5 *Arundo* plants displaying late symptoms (i.e., black streaks inside leaf sheath, Figure 1) of larval and pupal development of *L. donacis* from which we cut one 25 cm long internode with leaf sheaths. Once cut, the internodes were kept inside aerated plastic boxes in a fridge whenever possible until reaching EBCL laboratory. There, leaf sheaths were separated from each internode and the surfaces sterilized with 70% ethanol wipes. Under aseptic conditions, ~5 cm$^2$ of each leaf sheath was cut longitudinally. The inner parts of these longitudinal sections, i.e., the fungal communities, mainly black spores causing the black streaks, were scraped off using a flamed scalpel blade (Figure 2). All scrapings from each internode were pooled and suspended in 10 mL of sterile water.

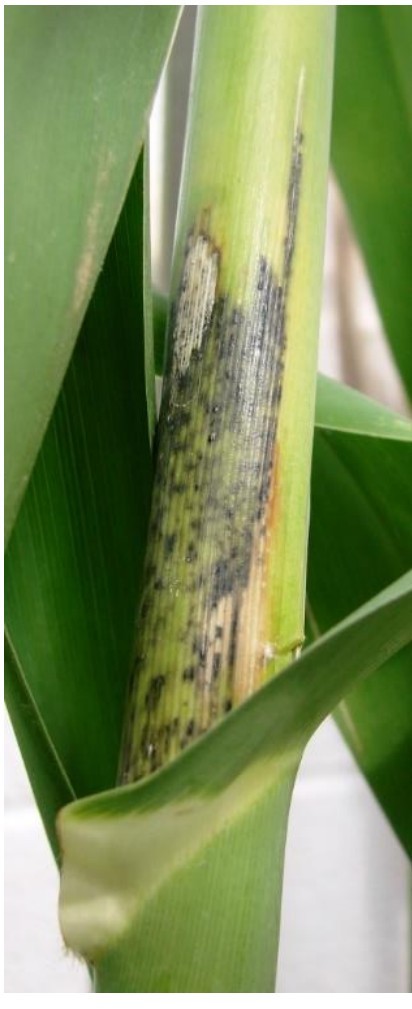

**Figure 1.** Late symptoms (i.e., black streaks inside leaf sheath of *A. donax*) of attacks by *L. donacis*.

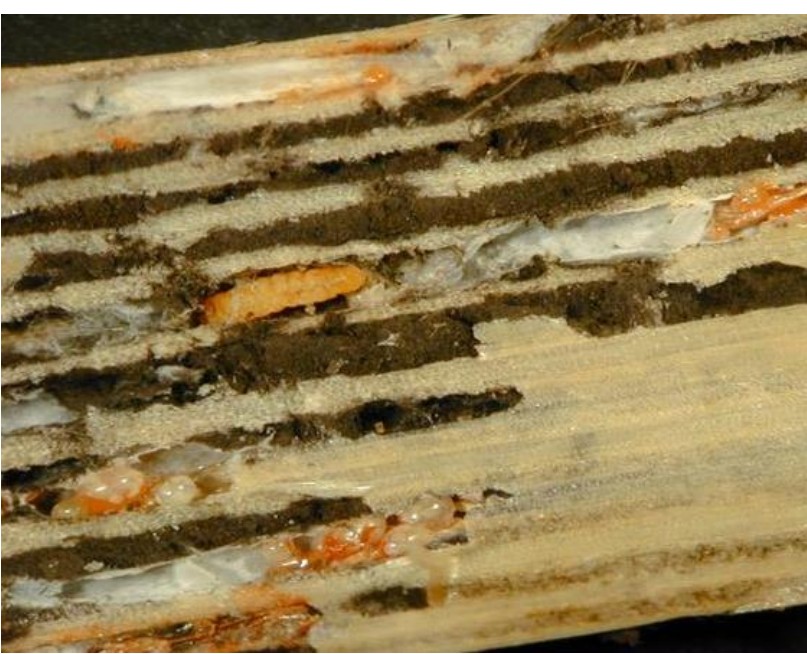

**Figure 2.** Fungal communities growing inside *A. donax*'s leaf sheath galleries mined by *L. donacis* larvae.

### 2.2. Fungal Communities Isolated from Leaf Sheath of Giant Reed in Texas

A total of 5 populations of *A. donax* were surveyed in 2012 in Texas along the Rio Grande where the grass does not display black streaks inside the leaf sheath because *L. donacis* is not present in this area, and there are no other leaf miners feeding on its leaf sheaths (Table S1). Harvested internodes with leaf sheaths (1 per plant, 3 plants per site) were treated as described above prior to transportation to the USDA-APHIS Biological Control Quarantine Laboratory, Edinburg, TX, USA. There, leaf sheaths were soaked in 5% sodium hypochlorite solution for 2 min, and then extensively rinsed in 3 baths of sterile water for 4 min. Under aseptic conditions, ~4 cm$^2$ of each leaf sheath was sliced longitudinally and placed directly onto agar media plates.

### 2.3. Fungal Isolation

In total, 120 scraping pools (24 populations × 5 plants) for the native range and 45 leaf sheath slices (5 populations × 3 plants × 3 leaf sheaths) for Texas were used for fungal isolation using a standard plating method. In total, 100 microliters of the homogenized pool of scrapings described above were distributed and leaf sheath segments (for Texas) were placed onto Potato Dextrose 1% Yeast Extract Agar (PDYA) supplemented with 0.5 g/L Chloramphenicol (Fluka™). All fungal colonies which developed within 14 days at 22 °C were transferred to PDYA media for single spore isolation and for preservation in EBCL or subsequently in Edinburg, TX culture collections. Subcultures on PDYA were roughly divided into different "phenotypes" according to colony color and texture on PDYA. When in each population, more than one fungal colony appeared phenotypically similar, then only one colony was selected for further molecular identification.

### 2.4. Conidia Associated with the Ovipositor and the Ovarioles of L. donacis Females

For 2 sites in the native range (Ireapetra in Crete and Prades le Lez in France, Table S1), a few Arundo internodes additional to those harvested for the fungal assemblage study were set up in Plexiglas boxes containing about 4 cm of moistened mixture (1/1) of sand/vermiculite in the laboratory quarantine. Internodes were placed inside insect proof cages maintained at 25 °C under 13 h of daylight. After several months, four females emerged from the canes harvested in Crete (Table S1), and each female was placed on one PDYA plate to observe potential oviposition holes with a Wild M400 macroscope (Heerbrugg, Switzerland). When oviposition holes were observed, the plate was incubated for

fungal colony development as described above. From the internodes harvested in France, 12 females emerged in May 2013 (Table S1), and they were collected freshly dead. A fungal isolation strategy different from the above was adopted. Under aseptic conditions, the ovipositor of six females was extracted using flamed forceps, dipped several times into one PDYA plate and incubated for fungal colony development as described above. The abdomens of the other six females were opened using a flamed scalpel and a flamed needle put in contact with the ovarioles. The needle was dipped several times into one PDYA plate and incubated for fungal colony development as described above.

### 2.5. DNA Extraction, PCR, and Sequencing

One agar slant of 1 cm² from each single spore culture was transferred aseptically into 250 mL Erlenmeyer Flasks containing 100 mL of 24 g/L Potato Dextrose Yeast Extract Broth (PDB). The flasks were incubated on a rotary shaker at 150 rpm, 25 °C for 3–4 days (Figure 3).

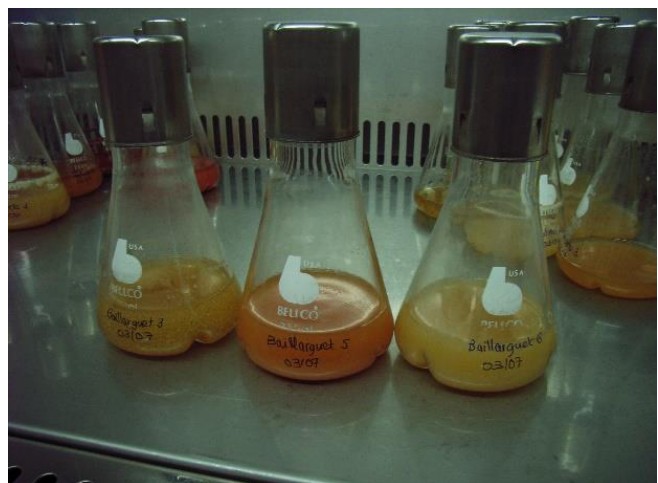

**Figure 3.** Fungal isolates in liquid cultures recovered from a *Lasioptera donacis'* fungal community sampled at Baillarguet in France. The three isolates belong to three different MOTUs.

Liquid cultures were centrifuged at 4000× *g* for 5 min at 4 °C. The resulting mat was rinsed with sterile 0.9% (wt./vol.) sodium chloride and centrifuged at 4000× *g* for 5 min at 4 °C. This procedure was repeated twice, and the mycelial mat was stored at −80 °C until used. Genomic DNA was extracted from ~100 mg of the mycelial mat using autoclaved glass beads (0.5 mm in diameter), mortar and pestle, and the DNeasy Plant mini-Kit (Qiagen, Courtaboeuf, France) following the manufacturer's protocol, with some variations to improve DNA extraction. Next, 1% (*w/v*) sodium bisulfite was added to the lysis buffer just before use, and the lysis was carried out at 65 °C for 20 min with regular shakings. To amplify internal transcribed spacer regions (ITS) of the nuclear ribosomal RNA operon, we targeted the primers ITS1F [26] and ITS4 [27] that are classically used for species identification in the two major phyla of fungi: *Ascomycota* and *Basidiomycota*. PCR was performed with a 48 µL reaction mixture and 2 µL of diluted DNA (up to 4 ng) in a Perkin Elmer 9700 Thermocycler (Perkin Elmer, Courtaboeuf, France). The reagent concentrations were 1 × PCR buffer (Qiagen), 1 Unit Qiagen *Taq* Polymerase, 200 µM dNTPs, and 0.3 µM of each primer. ITS amplifications were carried out as follows: initial denaturation at 94 °C for 3 min, followed by 35 cycles of denaturation at 94 °C for 30 s, annealing for 30 s at 50 °C, elongation for 1 min at 72 °C, and final extension at 72 °C for 10 min. To further resolve the relationship of *Apiospora arundinis* fungus isolated from *L. donacis* with other congeners, we amplified the β-tubulin gene (TUB2) and introns, and the 5′ portions of translation elongation factor (TEF1) 1α coding region and introns using the primers T1 [28] and Bt2b [29], and the primers EF1-728F [30] and EF2 [31], respectively,

using the same PCR conditions as described for ITS except for the annealing temperature, which was 52 °C. Both strands of all amplicons were sequenced. The chromatograms were analyzed with the BioEdit v7.1.3.0 program [32] and a consensus sequence obtained for each PCR product.

### 2.6. Sequence Data Analysis

The ITS sequences were analyzed using the CLC Microbial Genomics Module (version X) for de novo MOTU clustering. A value of 97% of ITS region identity was used as a DNA barcoding threshold for MOTU clustering [33,34]. A representative sequence of each MOTU was selected and searched using basic local alignment search tool (BLASTN) [35] against (1) the UNITE + INSD fungal ITS databases within the PlutoF web platform [36] and (2) the GenBank public sequence databases [37] by selecting the most reliable sequence as a reference (the sequences originated from mycologists or taxonomists, yielded from taxonomical or phylogenetical studies). The UNITE fungal Species Hypothesis (SH) at 1.5% threshold [38] was also added to each of the taxonomical assignments (Table 1). Some MOTUs were made of several amplicon sequence variants (ASVs) and all sequences representative of each MOTU have been deposited in GenBank with accession nos. OP970564 to OP970653 (Table 1). All ASVs which are representative of the within-MOTU variability were aligned using MUSCLE, and a Maximum Likelihood (ML) phylogenetic analysis was conducted using partial deletion of the gaps in MEGA X [39]. In this analysis, a Kimura 2-parameter model of substitution was employed with a gamma shape parameter of 0.90. Branch support was evaluated with 1000 bootstrap replicates. BLASTN was used to select *Apiospora arundinis* isolates for which ITS, TEF1, and TUB2 sequences could be retrieved from NCBI database. In addition, to each dataset of ITS, TEF1 and TUB2, we added one sequence acquired for the American *Apiospora arundinis* ATCC® 90177 strain Ar-21, which is currently used for host range testing of *L. donacis* in Texas [6]. Alignment for each marker was loaded in MEGA X for finding the best substitution model [39]. Models K2P [40] and JC [41] were selected for TUB2 and ITS + TEF1, respectively. The multigenic alignment of 1669 characters including 3 partitions (ITS, TUB2, and TEF1 with introns) was loaded in MrBayes 3.2 [42], where a Bayesian analysis was performed with data partitioned, 2 independent analyses of 4 Markov chains calculated for 1 million generations, and a 25% "burn-in". As the a priori outgroup for the phylogenetic analysis *Apiospora malaysiana* (Crous) Pintos and P. Alvarado, comb. nov. (Basionym: *Arthrinium malaysianium)* was chosen, which is relatively closely related to *A. arundinis* [43,44].

**Table 1.** List of the MOTUs evidenced in the study with taxonomic assignment provided by UNITE database, respective Blast similarity scores, and associated Genbank accession numbers. In bold are the two MOTUs recovered from ovarioles and ovipositors of *L. donacis*.

| MOTU Number/ Reference UNITE Fungal Species Hypothesis (SH) | Blast Similarity Score % (E-Value = 0) | Fungal Species/Class/Order | Genbank Accession Number |
|---|---|---|---|
| MOTU1/SH1541923.08FU | 99.4–100 | *Sarocladium terricola /* Sordariomycetes/ Hypocreales | OP970564 to OP970582 |
| MOTU2/SH1541921.08FU | 99.65 | *Sarocladium strictum /* Sordariomycetes/ Hypocreales | OP970583 & OP970584 |
| MOTU3/SH1541924.08FU | 99.65 | *Sarocladium spinificis /* Sordariomycetes/ Hypocreales | OP970585 to OP970587 |

| MOTU Number/ Reference UNITE Fungal Species Hypothesis (SH) | Blast Similarity Score % (E-Value = 0) | Fungal Species/Class/Order | Genbank Accession Number |
|---|---|---|---|
| MOTU4/SH1541928.08FU | 99.30 | *Nectria* sp./ Sordariomycetes/ Hypocreales | OP970588 & OP970589 |
| MOTU5/SH1567483.08FU | 99.62–100 | *Fusarium sporotrichioides*/ Sordariomycetes/ Hypocreales | OP970590 to OP970596 |
| **MOTU6/SH1610159.08FU** | **99.63–100** | ***Fusarium proliferatum*/ Sordariomycetes/ Hypocreales** | **OP970597 to OP970615** |
| MOTU7/SH1610157.08FU | 100 | *Fusarium* sp./ Sordariomycetes/ Hypocreales | OP970616 |
| MOTU8/SH1505921.08FU | 99.32 | *Alternaria* sp./ Sordariomycetes/ Hypocreales | OP970617 |
| MOTU9/SH1526398.08FU | 100 | *Alternaria alternata*/ Sordariomycetes/ Hypocreales | OP970618 to OP970621 |
| MOTU10/SH1507367.08FU | 100 | *Botryosphaeria dothidea*/ Dothideomycetes/ *Botryosphaeriales* | OP970622 to OP970625 |
| MOTU11/SH1615599.08FU | 100 | *Chaetomium globosum*/ Euascomycetes/ Sordariales | OP970626 |
| MOTU12/SH1572816.08FU | 100 | *Cladosporium cladosporioides*/ Dothideomycetes/ Capnodiales | OP970627 & OP970628 |
| MOTU13/SH1572792.08FU | 100 | *Cladosporium velox*/ Dothideomycetes/ Capnodiales | OP970629 & OP970630 |
| MOTU14/SH1572792.08FU | 100 | *Cladosporium aciculare*/ Dothideomycetes/ Capnodiales | OP970631 |
| MOTU15/SH1572820.08FU | 100 | *Cladosporium ramotenellum*/ Dothideomycetes/ Capnodiales | OP970632 |
| MOTU16/SH1526406.08FU | 100 | *Bipolaris* sp./ Dothideomycetes/ *Pleosporales* | OP970633 & OP970634 |
| MOTU17/SH1524421.08FU | 99.82 | *Lecanicillium lecanii*/ Sordariomycetes/ Hypocreales | OP970635 & OP970636 |
| MOTU18/SH1547057.08FU | 100 | *Epicoccum italicum*/ Dothideomycetes/ *Pleosporales* | OP970637 |
| MOTU19/SH1516144.08FU | 99.63–100 | *Penicilium funiculosus* syn: *Talaromyces funiculosus*/ Eurotiomycetes/Eurotiales | OP970638 to OP970644 |

**Table 1.** *Cont.*

| MOTU Number/ Reference UNITE Fungal Species Hypothesis (SH) | Blast Similarity Score % (E-Value = 0) | Fungal Species/Class/Order | Genbank Accession Number |
|---|---|---|---|
| MOTU20/SH1529984.08FU | 100 | *Penicilium rubens*/ /Eurotiomycetes/Eurotiales | OP970645 & OP970646 |
| MOTU21/SH1530001.08FU | 100 | *Penicilium terrigenum*/ /Eurotiomycetes/Eurotiales | OP970647 |
| MOTU22/SH1547057.08FU | 100 | *Phoma* sp./ Dothideomycetes/*Paleopoles* | OP970648 |
| **MOTU23/SH1540045.08FU** | **100** | ***Apiospora arundinis* syn:** ***Arthrinium arundinis*/** **Sordariomycetes/Xylariales** | **OP970649 to OP970653** **MF627422 [45]** |

## 3. Results

*3.1. Assemblage of Fungal Endophytes Inhabiting the Fungal Communities Associated with L. donacis and Leaf Sheath of Giant Reed*

All scrapings obtained from the fungal communities associated with *L. donacis* galleries in leaf sheaths sampled across the Mediterranean region and all sections of leaf sheaths sampled in Texas yielded cultivable fungal colonies. Some samples gave rise to multiple and diversely colored colonies (Figure 3). The grouping of these cultivable colonies based on their phenotype yielded 65 and 19 isolates in the Mediterranean region and in Texas, respectively. Unfortunately, the exact number of cultivable colonies obtained per plant and per site has not been recorded prior the phenotypic grouping; therefore, it is not possible to estimate the representativeness of these isolates in the original sampling. Using a threshold of 97% nucleotide similarity of the ITS sequences, all isolates were sorted into 22 different MOTUs (Table 1 and Table S2), of which 15 MOTUs were associated with the fungal community associated with *L. donacis*. ITS sequences provided enough resolution to identify all MOTUs at least to the genus level, and the phylogenetic relationships between all isolates is presented in Figure S1. The 22 MOTUs identified here belonged to 6 orders in the Ascomycota, i.e., Hypocreales, Eurotiales, Botryosphaeriales, Capnodiales, Pleosporales, and Sordariales, following the classification established by [46]. Hypocreales represented 70.77% (46/65) of all cultivable isolates obtained from scrapings from the native range and 68.42% (13/19) of all cultivable isolates sampled from leaf sheath slices in Texas, although its overrepresentation in the native range was not significant at the 0.05 level ($\chi^2 = 0.753$, $p = 0.385$). No particular MOTU was systematically recovered from the fungal communities regardless of the sites, though the cultivable mycota recovered from the fungal communities was largely dominated by *Fusarium* (35.38%, 23/65) and *Sarocladium* (30.76%; 20/65), both represented in Spain, France, Crete, and Greece, and to much less of an extent by *Penicilium* (13.8%; 9/65). The *Sarocladium* and the *Fusarium* genera were particularly dominated by *Sarocladium terricola* (J.H. Mill., Giddens and A.A. Foster) A. Giraldo, Gené and Guarro (90%, 18/20) and *Fusarium proliferatum* (Matsushima) Nirenberg (69.56%, 16/23), respectively. The fungal assemblage associated with giant reed leaf sheath in Texas was mostly represented by *Sarocladium* (21.05%, 4/19), *Alternaria* (21.05%, 4/19), and *Fusarium* (15.7%, 3/19). Among the twenty-two MOTUs evidenced in this study, four were common to the two ranges, including *S. terricola* and *F. proliferatum* (Table 1). Eleven MOTUs were unique to *L donacis* galleries and seven to the giant reeds in Texas. Among these eleven MOTUs unique to the fungal communities associated to *L. donacis* galleries, six were found in only one site in Spain and France in the Mediterranean region.

*3.2. Lasioptera Donacis Is Carrying Endophyte Conidiae*

Fungal conidiae collected from ovipositors and ovarioles of *L. donacis* yielded cultivable colonies, and only one colony per ovariole or ovipositor. Conidiae associated

with ovipositors and ovarioles were found to belong to two MOTUs. One (MOTU 6) was previously evidenced from the scrapings, i.e., *F. proliferatum* (*Matsush.*) *Nirenberg* found in females collected in Crete, and a new one was found in females collected in France, MOTU 23, i.e., *Arthrinum arundinis* which has been recently synonymized with *Apiospora arundinis* (Corda) Pintos and P. Alvarado comb. nov. [44] (Table 1 and Table S2). *Arthrinium arundinis* and its synonyms have been reported in 74 different plant hosts [47] and are commonly occurring in different habitats or biotopes [47,48]. A phylogeny of a combined ITS, TUB2, and TEF1 dataset between the MOTU 23, the American ATCC strain Ar-21 used in host range testing in Texas [6], and all *A. arundinis* retrieved from GenBank is presented in Figure 4. The phylogeny placed the MOTU 23 within a well-supported clade which included also *A. arundinis* cultures from Poaceae (CBS 464.83, which originated from dead culms of Phragmites australis in Netherland [47] and LC4493 and LC 7160 which originated from bamboo leaf in China) [49] (Figure 4). Sequences of the American ATCC strain Ar-21 were found similar to those of the CBS strain 114316 also isolated from Hordeum vulgare, but in Iran, and both strains clustered in the other well-supported clade.

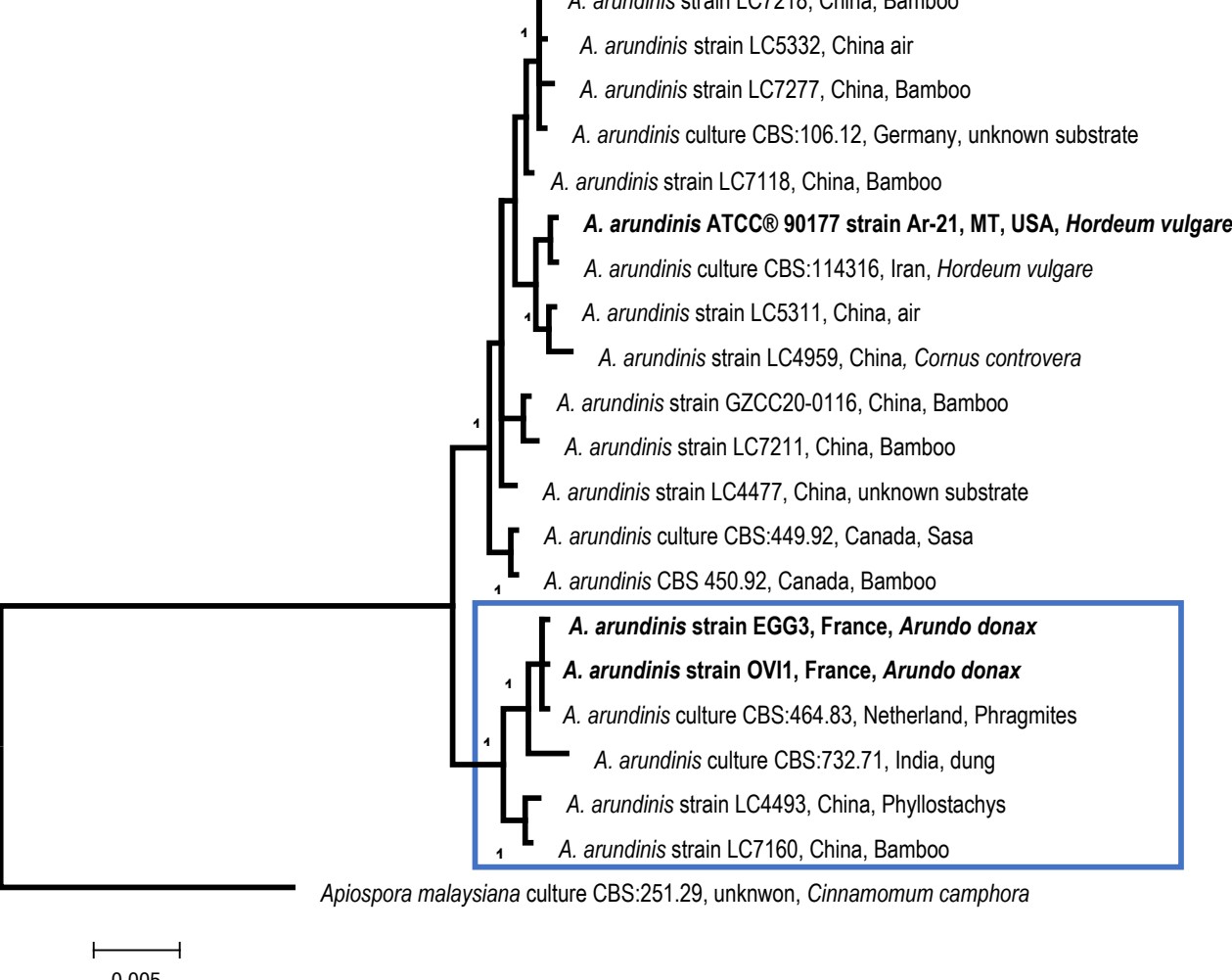

**Figure 4.** A 50% majority rule consensus phylogram obtained in MrBayes after the analysis of combined ITS rDNA, TUB2, and TEF1 sequences (introns included) of *Apiospora arundinis*. Nodes were annotated if supported by >0.95 Bayesian posterior probabilities. Bold names represent samples sequenced in the present study.

## 4. Discussion

The 65 fungal colonies isolated from the communities associated with *L. donacis* across a large biogeographic range were successfully identified at least at the genus level. The examination of this mycobiota neither evidenced *Aspergillus niger* which was thought to be associated with *L. donacis* by Coutin [19], nor any other fungal species associated with other *Lasioptera* species such as *Sporothrix* sp., *Ramichloridum subulatum*, and *Aureobasidium pullulans* as reviewed by Rohfritsch [14–16]. In each site across this range, we never recovered systematically the same fungal taxa but barely a similar fungal assemblage. The mycobiota recovered from *L. donacis* galleries was clearly dominated by taxa belonging to Hypocreales within the Ascomycota, a characteristic that is consistent with an assemblage of non-systemic endophytes as is often observed in temperate grasses [50–52]. Nonetheless, fungal endophytism is not a stably trophic state, but is instead a transient trophic mode. Many fungi may evolve their lifestyles as saprophyte–endophyte pathogens to adapt to various changes in host and environmental conditions [53,54]. Two MOTUs, *Fusarium proliferatum* and *Sarocladium terricola*, predominated in this fungal assemblage and were found to co-occur in some sites, i.e., Agia Triada in Greece, and Ireapetra in Crete. Interestingly, *F. proliferatum* was one MOTU from the ovipositor of the *L. donacis* females collected in Ireapetra in Crete. *F. proliferatum* is not only a fungus with a worldwide distribution that has been associated with a variety of diseases in important economical plants, including corn and bananas, but also as an opportunist or an entomopathogen of different insect species as mango leafhoppers [55], the sugar cane scale, the brown planthopper, the plum curculio [56], and the chestnut gall wasp [57]. *Apiospora arundinis* (Corda), (syn: *Arthrinium arundinis*) was the second MOTU isolated from the ovarioles and ovipositors of *L. donacis* females collected in France. *Arthrinium arundinis*, which was first described by [58], belongs to a widespread and ecologically diverse genus. *Arthrinium* commonly occurs as a saprobe on grasses of Poaceae, and also on leaves, stems, and roots of a range of different plant substrates [43,47,48]. *Arthrinium* has been reported as a plant pathogen, with *A. arundinis* causing kernel blight of barley [59], and as an endophyte in plant tissue producing secondary metabolites with antimicrobial properties [60]. We did not see evidence of *A. arundinis* in the *Lasioptera* galleries from *A. donax* in the Mediterranean region or leaf sheath of *A. donax* in Texas; to our knowledge, there is no report in the literature of the occurrence of *A. arundinis* in *A. donax* [47]. However, at least four *Arthrinum* species, *A. ibericum*, *A. italicum*, *A. marii*, and *A. phragmitis*, were found growing in *A. donax* in the Mediterranean biogeographical region [43], so we still could hypothesize that *A. arundinis* might be present in this grass but not sampled. Our sampling methods may not have revealed fungi that occur at the beginning of the infection process such as *Arthrinium* spp. that have been recovered from the ovipositor of *L. donacis* or that are difficult to recover using our culture-dependent method. Our methods may have been biased toward fungal associates found in the later stages of leaf sheath decomposition. We hypothesized that *A arundinis* may be the first fungus to colonize the oviposition wound created by *L. donacis* females and then a mix of secondary saprophytes/endophytes emerge. This finding has been the working hypothesis that *A. arundinis* was associated with oviposition and larval feeding of *L. donacis* [6,7,21,61]. The growth of these fungi in leaf sheaths with mature *L. donacis* larvae may obscure the presence of *A. arundinis* and explained the absence of *A. arundinis* in our sampling. Isolates of *Sarocladium* dominated the fungal assemblages in our present investigation. *Sarocladium* spp. are saprophytes found in soil and in association with plants, especially members of the Poaceae, including weed grass species [62] but also common endophytes of grasses [63,64]. Endophytic *Sarocladium* spp. may have protective roles on grass hosts against abiotic stresses and pathogens [62]. Endophytic isolates of *S. terricola* from *Brachiaria* spp. were found to produce diffusible metabolites that are toxic to plant pathogens [63,64]. Cultivable mycobiota lining the galleries of *L. donacis* was then compared with the endophytic assemblage present in the giant reeds in Texas where the insect does not occur. The latter was clearly dominated by taxa belonging to Hypocreales within the Ascomycota, a characteristic that is consistent with an assemblage of non-systemic

endophytes as often observed in temperate grasses [50,51]. *Sarocladium, Fusarium*, and *Alternaria* predominated in this fungal assemblage. Interestingly, *Sarocladium* was also the predominant phylotype that was evidenced in pupae of *L. donacis* [45]. *Alternaria* as well as two less commonly detected genera, *Epicoccum* and *Cladosporium,* were previously evidenced in *A. donax* growing on gypsum and saline soils in Central Spain [65]. Several endophytes/saprophytes assigned to five MOTUs were found both in *L. donacis'* galleries and in *A. donax* in Texas. Assuming that endophytes/saprophytes may play a role in the biology of *L. donacis,* one role suggested by [19] pertains to a potential sapromycophagous feeding habit of the midge, similar to *Lasioptera rubi* Reeger in *Rubus* spp. host plants [13]. Some of these endophytes/saprophytes fortuitously will already be present in invasive stands of *A. donax* when this midge will be released in Texas. There could be ecological substitutes for their European congeners. If this is the case, then European fungi would not need to be imported or released with *L. donacis*. It is clear that *A. donax* in Texas harbors a variety of fungal endophytes within its leaves. Antagonistic effects of endophytes on plant pathogens have been extensively noted in a variety of host plants [66], and recently on the rust *Puccinia komarovii var. glanduliferae*, introduced into the UK for biological control of the invasive weed Impatiens *glandulifera* [67]. Additionally, it is known that the chemical changes induced by endophytes are systemic and have detrimental effects on some insect herbivores leading to their recent description as "plant bodyguards" [68]. These effects are particularly noticeable for insects feeding on phloem, in which many secondary metabolites are carried. In the *Cameraria* sp. leaf miner-*Quercus emoryi* system and the associated fungal endophytes, Wilson and Stanley [69] found that leaf miner ovipositional preferences may have been selected by avoidance of fungal endophytes and associated endophyte-mediated antagonism. Nothing is known about the nature of the relationships between endophyte/saprophyte fungi living in *A. donax*, other than *A. arundinis* and *L. donacis*, and this needs to be addressed in support of the introduction of *L. donacis* as a biocontrol agent of *A. donax* in Texas. The fact that *Sarocladium* and *Fusarium* are shown to be widespread in the fungal community associated with *L. donacis* in the present study, coupled with vertical transmission in pupae of *L. donacis* in the case of *Sarocladium* [45], and ovarioles in the case of *Fusarium* in the present study, led to the new hypothesis that *Sarocladium* and *Fusarium* could likely be important in the biology of the midge either as direct food sources or pre-digestion of plant material. Furthermore, the role played by *A. arundinis* in this multitrophic system is not exclusive, contrary to what has been suggested in previous studies [6,21,61].

In conclusion, this study has shown that the fungal community lining the galleries mined in the giant reed by *L. donacis* are mainly composed of an assemblage of non-systemic endophytes, but not of a primary fungal symbiont as suggested in the literature. This first study which identified numerous isolated taxa from a culture-dependent method to species level, lays the groundwork for future fungal community studies. As one main limitation of the culture-dependent approach is that unculturable species and some slow growing or weakly competitive species may not be isolated, future endophyte research should benefit from advances in culture-independent approaches such as next generation sequencing (NGS). These methods infer the composition of endophytic communities at an astounding level of information for a better understanding of their ecology and distribution [70].

**Supplementary Materials:** The following supporting information can be downloaded at https://www.mdpi.com/article/10.3390/d15040571/s1, Table S1: Collection sites and dates, name of the collector, and abbreviation for the fungus isolate recovered in each site; Figure S1: Circular phylogenetic tree inferred from maximum likelihood (ML) using ITS-Amplicon variant sequences depicting the within and between MOTUs diversity of the endophytes inhabiting *Lasioptera donacis* galleries and leaf sheath of giant reed; Table S2: List of the MOTUs evidenced in the study with taxonomic code assignment provided by UNITE database and associated strains.

**Author Contributions:** Conceptualization, M.-C.B., J.A.G., G.M., M.C. and A.K.; methodology, M.-C.B., F.G., G.M., J.K., M.C. and A.T.V.; software, M.-C.B.; validation, M.-C.B., F.G. and J.A.G.; writing—original draft preparation, M.-C.B. and J.A.G.; writing—review and editing, M.-C.B., J.A.G., G.M., F.G., J.K., M.C. and A.K. All authors have read and agreed to the published version of the manuscript.

**Funding:** This research received no external funding.

**Institutional Review Board Statement:** Not applicable.

**Data Availability Statement:** The data and alignments supporting the current study are available from the first author (Marie-Claude Bon; mcbon@ars-ebcl.org) on request.

**Acknowledgments:** We would like to thank Crystal Salinas for her great help in isolate preparation in Texas and Marie-Line Merg for her technical support during her undergraduate internship in France, Patrick Moran (USDA-ARS) for his careful reading of our manuscript and many insightful comments and suggestions. Mention of trade names or commercial products in this publication is solely for the purpose of providing specific information and does not imply recommendation or endorsement by the USDA; USDA is an equal opportunity provider and employer.

**Conflicts of Interest:** The authors declare no conflict of interest.

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
