# Peer review of "Detection of a Diverse Endophyte Assemblage within Fungal Communities Associated with the Arundo Leaf Miner, Lasioptera donacis (Diptera: Cecidomyiidae)"

_diversity, doi:10.3390/d15040571_

Round 1

Reviewer 1 Report

The authors describe the endophytic fungal assemblages  associated with the Arundo leaf miner, Lasioptera donacis.

This is valuable and informative. Design and execution are valuable and informative.

 Some remarks:

The term “so-called” is too colloquial. Putative is more scientifically accurate.

The term ambrosia is normally used to describe ambrosia fungal symbionts associated with insects with special features that carry these fungi (mycangia), i.e., ambrosia beetles, subfamilies Scolytinae and Platypodinae (Coleoptera, Curculionidae). Google scholar shows one single paper referring to this term and the midge Lasioptera donacis, which is a diptera.

I would strongly suggest the authors to substantiate they association ambrosia fungi – flies (which they do for one fungus) but to link this with the common ambrosia -insect known associations, i.e., Curculionidae. It is a big statement to go from Ambrosia fungi studied since the 1960’s in Cuurculionidae (formerly Scolytidae), to Diptera. Ambrosia is linked to a complete life cycle with curculionidae, mycangia, vectors, symbiotic relationships, etc. This is not proven for the Diptera in question. There is one association reported with one fungus, which the authors mention but there is no reference for it : line 53-54 "In the most-documented cases, the gall midge species has specialized relationships with one fungal species". No reference.

Hence I would use the term putative instead of ‘so-called” and encourage authors to corroborate the use of Ambrosia with Diptera. The information is mostly historical. No mention of Ambrosia beetles. A sentence stating “we adopt the term Ambrosia, to describe…..as similarities have been documented. These similarities are :….”  Something like this.

I would not use the term actually, since it is a big jump. I would use  fungal community assemblages. But if the authors can substantiate better the use of the term this cold potentially be done. I have doubts though.

I honestly thought the species listed in the title was an ambrosia beetle. I am not sure if the use of this term is correct given we are dealing with midges flies (Diptera: Cecidomyiidae). There is a substantial historical information on the term ambrosia in the introduction but the common use of the term in invertebrate pathology, i.e., ambrosia beetles, is not mentioned. This should be mentioned. In addition the order and family should be listed in the title given the information above.

I am assuming the primers were universal primers for the target regions and fungi?

If so please list that so we know the strategy that was taken to select primers as this is a critical step in the project.

For some MOTUs, different ITS sequences have  been obtained in different isolates. I am not sure what this means. Different MOTUs or different ASVs? (amplicon sequence variants). Different ITS sequences are expected as this is a hypervariable region in fungi. Hence, I assume authors mean ASVs but not sure as they state “for some MOTUs”

 Line 227 to 238 is all in italic. 263 to 268 as well.

Author Response

The term “so-called” is too colloquial. Putative is more scientifically accurate.The term ambrosia is normally used to describe ambrosia fungal symbionts associated with insects with special features that carry these fungi (mycangia), i.e., ambrosia beetles, subfamilies Scolytinae and Platypodinae (Coleoptera, Curculionidae). Google scholar shows one single paper referring to this term and the midge Lasioptera donacis, which is a diptera.I would strongly suggest the authors to substantiate they association ambrosia fungi – flies (which they do for one fungus) but to link this with the common ambrosia -insect known associations, i.e., Curculionidae. It is a big statement to go from Ambrosia fungi studied since the 1960’s in Cuurculionidae (formerly Scolytidae), to Diptera. Ambrosia is linked to a complete life cycle with curculionidae, mycangia, vectors, symbiotic relationships, etc. This is not proven for the Diptera in question. There is one association reported with one fungus, which the authors mention but there is no reference for it : line 53-54 "In the most-documented cases, the gall midge species has specialized relationships with one fungal species". No reference.Hence I would use the term putative instead of ‘so-called” and encourage authors to corroborate the use of Ambrosia with Diptera. The information is mostly historical. No mention of Ambrosia beetles. A sentence stating “we adopt the term Ambrosia, to describe…..as similarities have been documented. These similarities are :….”  Something like this.I would not use the term actually, since it is a big jump. I would use  fungal community assemblages. But if the authors can substantiate better the use of the term this cold potentially be done. I have doubts though.I honestly thought the species listed in the title was an ambrosia beetle. I am not sure if the use of this term is correct given we are dealing with midges flies (Diptera: Cecidomyiidae). There is a substantial historical information on the term ambrosia in the introduction but the common use of the term in invertebrate pathology, i.e., ambrosia beetles, is not mentioned. This should be mentioned. In addition the order and family should be listed in the title given the information above.

Thank you very much for addressing this issue. After a last search through the contemporary literature, it is clear that the term of Ambrosia, even putative Ambrosia should be restricted to beetles, mainly belonging to the subfamilies of Scolytinae and Platypodinae. In the taxonomic and biological description of the midge flies, in particular in Gagné and Jaschhof (2022), we could not find all the characteristics of the insects associated with Ambrosia fungi as described in Ambrosia beetles. To avoid perpetuating this historical confusion, we remove the term Ambrosia from the manuscript and replace it by fungal communities. We have also added few sentences in the introduction ( lines 52 to 58) and one reference to tentatively explain why we are no longer referring to so called Ambrosia used in the past literature.

I am assuming the primers were universal primers for the target regions and fungi? If so please list that so we know the strategy that was taken to select primers as this is a critical step in the project.

Yes, they are. We have modified the lines 179 to 182 and mentioned that these primers are classically used for species identification in the two major phyla of fungi: Ascomycota and Basidiomycota as mentioned by Gardes et al (1993) and White et al (1990). Based on our own experience, this primer set is the most efficient to amplify across these two phyla. All DNAs were amplified with success with this primer set.

For some MOTUs, different ITS sequences have  been obtained in different isolates. I am not sure what this means. Different MOTUs or different ASVs? (amplicon sequence variants). Different ITS sequences are expected as this is a hypervariable region in fungi. Hence, I assume authors mean ASVs but not sure as they state “for some MOTUs”

We have changed the lines 205 to 208 to take into consideration your remark. Indeed a MOTU could be made of one unique sequence or because of the threshold-based clustering algorithm could include several variants due to substitutions or indels. We have deposited all amplicon variant sequences in Genbank

 Line 227 to 238 is all in italic. 263 to 268 as well.

We have corrected this. Thank you

Reviewer 2 Report

The manuscript entitled "The so-called ambrosia of the Arundo leaf miner, Lasioptera donacis, harbors a diverse endophytic fungal assemblage" presents and interesting research related to microbial-insects interactions.

There are some suggestions that will improve the current form of the manuscript.

Abstract is clear and well written

Introduction section

The lines 75-95 should be presented as a separate paragraph and divided in two parts - The aim and objectives clearly stated, without any references. All the information linked to international literature presented separate. This will help the reader to understand what the manuscript will present. 

Materials and Methods are explicit and present well the resources and procedures of the experiment carried out by the authors.

Results section

The text related to the results obtained by authors have an appropriate length. The results are presented well and point the main findings of the research.

Table 1. Please find a condensed form of presentation. It is too long and split the text.

Discussion section is linked to international studies in the field and connect the results with similar studies.

Overall, the manuscript is interesting both from a methodological and a future research perspectives. 

Author Response

The manuscript entitled "The so-called ambrosia of the Arundo leaf miner, Lasioptera donacis, harbors a diverse endophytic fungal assemblage" presents and interesting research related to microbial-insects interactions.

There are some suggestions that will improve the current form of the manuscript.

Abstract is clear and well written

Introduction section

The lines 75-95 should be presented as a separate paragraph and divided in two parts - The aim and objectives clearly stated, without any references. All the information linked to international literature presented separate. This will help the reader to understand what the manuscript will present. 

Thank you very much for this remark. We have reorganized the introduction in order to evidence different paragraphs

Materials and Methods are explicit and present well the resources and procedures of the experiment carried out by the authors.

Results section

The text related to the results obtained by authors have an appropriate length. The results are presented well and point the main findings of the research.

Table 1. Please find a condensed form of presentation. It is too long and split the text.

Thank you for this remark. It has been quite tedious to reorganize the table 1 in a more condensed form. The only option was to delete the column associated with the names of the strains and to create an additional table (Table S2) with this information in association with their corresponding MOTU.

Discussion section is linked to international studies in the field and connect the results with similar studies.

Overall, the manuscript is interesting both from a methodological and a future research perspectives. 

thank you for your valuable remarks

Round 2

Reviewer 1 Report

The authors thoroughly revised the manuscript as per reviewers suggestions. I would just advise to revise this sentence as it had some grammatical issues. Other than that I think it is ready for publication. Thank you   " As Ambrosia fungi are known, in the vast majority of scientific literature, to be exclusively associated with members of beetles of the sub-families Scolytinae and Platypodinae (Coleoptera: Curculionidae), but not with Dipterans to our knowledge [17], we adopt the term fungal communities  when it comes to fungi that either directly or indirectly mediate the nutritional interaction of midge larvae with host plants."